# Multi-Scenario Pricing for Hotel Revenue Management

Submission Id: 239*

## ABSTRACT

Dynamic pricing algorithms have been widely studied to manage hotel and platform revenue over online travel platforms (OTPs). For better dynamic pricing, the accurate estimation of the market demand and the market competitiveness are crucial. However, the existing approaches obtain a pricing strategy tailored to each specific scenario using data only from that scenario. They are not considering the shared information between different scenarios, i.e., the data from different scenarios are not fully utilized. So we propose a Multi Scenario Pricing model (MSP) with a novel sharing structure design that leverages cross-scenario and specific information to capture more accurate market demand and competitiveness. Specifically, the model structure explicitly separates information into shared components as market demand and specific information as scenario-wise price competitiveness to prevent domain seesaw. To capture the inherent correlation between listings in different scenarios, an attention network named Price Competitiveness Representation Extraction (PCRE) is well-designed. Meanwhile, traditional metrics are skewed towards model that tends to reduce the price regardless of sample distribution. Thus we propose new offline evaluation metrics that shift attention with sample distribution to avoid biased pricing strategies, which is proved to be more closely related to actual business revenue. Our proposed MSP shows superiority under both offline and online experiments on real-world datasets. The multi-scenario industry dataset [1] and our code[2] are available. To the best of our knowledge, it will be the first real-industry multi-scenario pricing data.

## CCS CONCEPTS

• **Computing methodologies** → **Neural networks**; • **Applied computing** → **Forecasting**.

## KEYWORDS

Dynamic Pricing, Deep Learning, Multiple Scenario, Revenue Management

ACM Reference Format:
Anonymous Author(s). 2023. Multi-Scenario Pricing for Hotel Revenue Management. In *Proceedings of The Web Conference 2024 (WWW'24)*. ACM, New York, NY, USA, 9 pages. https://doi.org/XXXXXXX.XXXXXXX

[1]https://tianchi.aliyun.com/dataset/159383

[2]https://anonymous.4open.science/r/Multi-Scenario-Pricing-8E75/README.md

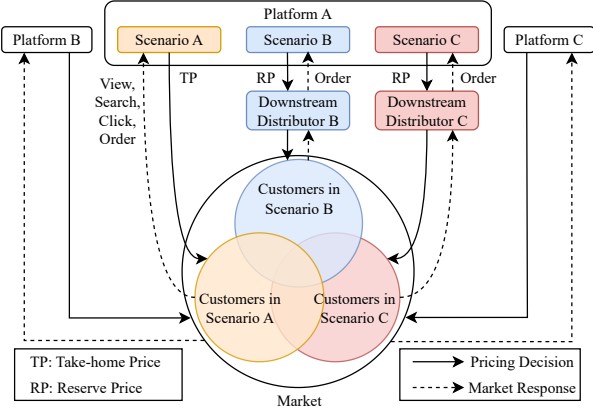

**Figure 1: Intuition of Three Scenarios.**

## 1 INTRODUCTION

With the explosive growth of online information and services, many online travel platforms (OTPs) have developed a multi-scenario sales mode to increase exposure and improve brand awareness so as to reach potential customers and cover more market demand. Platform A is an online travel platform with three distribution scenarios, where customers can query about listed prices of each room type on different check-in dates (referred to as listing). And listed prices are determined by dynamic pricing algorithms. All these scenarios share market demand, hotel characteristics, and contextual environment while having different pricing power and targeting customers with different consumption habits. On the one hand, in scenario A, we make a complete price decision on the take-home price that is directly shown to customers, while in scenario B and C, facing downstream distributors, we give a relatively lower reserve price so that distributor can have their own price-raising strategy over it. On the other hand, customer portraits vary with the scenario. While scenario A and C are more general, scenario B is attached to a map app. So scenario B naturally attracts consumers on the journey, who are inclined to choose hotels near airports and railway stations. Besides, data reflects market demand including historical search, view, click and order is available only for scenario A, as for scenario B and C, we receive order information from downstream distributors as summarized in Figure 1. In this paper, we are going to consider the problem in multi-scenario pricing.

As a lever to better match supply and demand, pricing strategy has been a pivotal tool in managing the overall revenue of OTPs. Traditionally, the pricing models are usually developed respectively under each certain scenario, but not multi-scenario. However, under the multi-scenario distribution mode, the traditional dynamic pricing approach that solely considers its own data for each scenario would lead to many challenges: 1) Hard to utilize the shared knowledge across scenarios like market demand especially for some

scenarios with sparse data, considering that there exists overlapping items and listings with similar characteristics in multiple scenarios. Specifically, in our pricing problem, historical exposure data, including user view, click, and search is only available in scenario A. 2) If we feed all the data into one model, the differences between scenarios would be completely ignored and all the data would be projected in the same feature space despite data heterogeneity in different scenarios, which easily causes the domain seesaw problem [28], especially considering that price competitiveness varies significantly between scenarios according to various pricing power and customer portraits. 3) Training a pricing model solely on its own data for each scenario can lead to considerably high calculation and maintenance costs as the number of pricing scenarios increases.

To tackle the above issues, we propose a Multi-Scenario Pricing model (MSP) with progressive demand extraction layers to utilize demand information across scenarios and scenario-wise attention network that explicitly separates scenario specific price competitiveness information. In Demand Representation Extraction (DRE), through the shared expert and multi-level structure, the information of multiple scenarios can be integrated to learn the market demand more accurately. With attention networks in the Price Competitiveness Representation Extraction module (PCRE), this model measures the similarity of hotels and temporal price fluctuation to enhance the learning process of price competitiveness. In this manner, critical pricing information, including demand and competitiveness, is well extracted from the multi-scenario dataset, which helps to comprehensively make a rational pricing suggestion.

Our main contributions are summarized as follows:

- We propose a novel Multi-Scenario Pricing model (MSP), which explicitly separates extraction modules to capture shared demand information and scenario-wise price competitiveness for better modeling complex correlation between scenarios and listings and addresses the negative transfer further to prevent domain seesaw problem. Extensive experiments on large-scale industrial data and online A/B tests show that our proposed method significantly outperforms existing methods both on multi-scenario pricing and single-scenario pricing problem.
- We propose a more comprehensive pricing metric that shifts attention according to sample distribution to accommodate class imbalance, which is well correlated with the online business metrics.
- A large-scale production dataset on hotel pricing collected from the online travel platform A is released with this paper. To the best of our knowledge, this is the first industrial production dataset with multi-scenario pricing. We hope this could help facilitate future research in dynamic pricing.

## 2 RELATED WORK

Existing approaches for dynamic pricing can be categorized into the following two categories: 1) model-driven approaches, 2) data-driven approaches.

Model-driven approaches perform revenue management based on the estimated demand function with explicit parameters [2, 3, 5, 6, 10, 11, 23]. For instance, Aviv and Pazgal [3] study the finite horizon dynamic pricing problem for the perishable products. They assume the prior distribution of the intensity is the gamma distribution, which is a conjugate distribution for the Poisson demand process. But they consider the demand function from is known. Şen and Zhang [23] try to resolve uncertainty about the demand function. They haven't assumed the function form of demand function, but also assume demand function come from a family of functions. The former approaches are trying to resolve the dynamic pricing problem in the finite horizon setting for the perishable products. For the perishable products, a fixed parameter to model the market condition is well suited with the selling season. But for the nonperishable products, the market condition would change over time. To overcome this issue, Araman and Caldentey [2] model the distribution of low and high market size with two-point prior. However, these priors may not meet with the real-world scenarios.

Data-driven approaches are utilizing the machine learning techniques to predict the optimal price based on the multiple factors that influence the demand [7, 8, 16, 18, 19, 21, 29, 30, 33, 34]. Rana and Oliveira [22] formulate the dynamic pricing problem as a discrete finite horizon Markov Decision Process (MDP) and use model-free reinforcement learning technique to learn an optimal pricing policy for maximizing revenue. AmalNick and Qorbanian [1] utilizing the wavelet neural network to predict the future demand, and combining the evolutionary algorithms to obtain the optimal pricing policy. Ye et al. [31] utilizing Gradient Boosting Machine (GBM) [12] to predict the booking probability of listing night and regress the optimal price with the model which is guiding by customized loss function. Zhang et al. [33] propose a novel sequence learning model which integrates DeepFM [13] and the seq2seq model [26] for predicting occupancy and predict the suggested price with a DNN model. Zhu et al. [34] proposed a multi-task learning procedure to tackle the data sparseness, which can provide a more robustness of occupancy prediction. Mussi et al. [20] consider the volume discounts setting. They estimate the demand curve to retrieves the optimal average price, then obtain the discounts for each volume threshold. However, they all ignore the shared information across multiple scenarios.

## 3 OUR PROPOSED METHOD

In this section, we introduce the pricing strategy and technical details of our proposed MSP. Since there are no accurate labels for pricing problem, we define listings with room nights higher than the past-month average room nights as **good-day listings**.

As discussed in Section 1, one general accurate demand curve that models the correlation between price and occupancy is hard to be extracted from different scenarios and easily causes domain seesaw. To overcome this, inspired by Ye et al. [31], we divide the solving process into two steps to model the highly non-linear relation between market value and product features.

First, we build a progressive extraction model with attention network that well extracts shared demand over the market and specific scenario price competitiveness. Next, tower modules learn the mapping from extracted information to **good-day probability** as well as parameters of the price suggestion function. Finally, the good-day probability is mapped to **price suggestion** via a non-linear price suggestion function [31] as the model output. Accordingly, we design the pricing network model as below.

## 3.1 Problem Formula

We define the multi-scenario pricing problem as $v_i^j = f_i^j(\mathbf{x}_i^j, p_i^j)$, where $i$ denotes the $i$-th scenario and $j$ denotes a specific listing, $v_i^j$ is the price adjustment ratio for listing $j$ under scenario $i$ given the current calendar price $p_i^j$, $\mathbf{x}_i^j$ contains all the other raw features of listing $j$ in scenario $i$ in following categories:

- hotel profile $\mathbf{x_p}^j \in \mathbb{R}^m$: brand, rating, hotel star, location, business district, bed size, etc.
- contextual features $\mathbf{x_t} \in \mathbb{R}^n$: week-of-year, event, holiday-or-not, etc.
- demand features $\mathbf{X}_{di}^j \in \mathbb{R}^{u \times t}$ in scenario $i$: historical clicks, search, uv, ipv, order nights, etc. ($u$: categories of demand sequences, $t$: length of time sequence)
- competitiveness features $\mathbf{X}_{ci}^j \in \mathbb{R}^{r \times t}$ in scenario $i$: ratios of the current listed price $p_i^j$ to the competitive price, average historical order payment, similar hotels order payment, etc. ($r$: categories of price ratio sequences, $t$: length of time sequence)

## 3.2 Network Structure

### 3.2.1 Good-day probability prediction.
Market demand experience would change with seasonality and events, transferring across the whole supply chain network. However, the transmission usually has a time lag between different scenarios. Sharing of demand information can tackle the scenario-wise information missing problem and enhance the accuracy and efficiency of information-lagging scenario pricing decisions. In the meanwhile, user portraits in different scenarios distinguish from each other, which results in distinguishing price competitiveness in different scenarios.

Considering this, we explicitly decouple the scenario-sharing and scenario-specific information into demand and price competitiveness and correspondingly decompose our model into two sub-modules as illustrated in Fig 2: Demand Representation Extraction (DRE) and Price Competitive Representation Extraction (PCRE).

• **Demand Representation Extraction (DRE)** This module plays the role of transferring and sharing valuable demand information as well as extracting specific scenario demand. To address the domain seesaw problem, in DRE, shown as the yellow part in Fig. 2. We employ the Progressive Layered Extraction (PLE) with multiple-layer information extraction structure inspired by [27], which allows the model to learn more complex demand correlation between scenarios progressively. Each layer is composed of scenario-shared expert and scenario-specific experts that explicitly separates shared demand information and scenario-specific information of the current scenario. Since the following discussion all takes listing $j$ as example, the superscripts $j$ are omitted for the sake of formula brevity. Given a listing $j$ in scenario $i$ with demand features $\mathbf{X}_{di}^j \in \mathbb{R}^{u \times t}$. Firstly, we generate demand embedding $Emb_{di} \in \mathbb{R}^u$ with Gate Recurrent Unit (GRU) to capture the temporal correlation [17]. And then, scenario-specific expert $Expert_i$ takes its corresponding scenario demand embedding $Emb_{di}$ as input, while the scenario-shared expert absorb knowledge from all scenario demand embeddings. Which means the parameters of scenario-specific experts are only related to the corresponding

scenario demand while the shared experts are related to all the scenarios.

To learn the complex demand information progressively, DRE is composed of multiple similar extraction layers. For brevity, we take the $l$-th extraction layer as an example to demonstrate the structure details. The input are the demand representations $D_i^{l-1}$ which are extracted by the former layer ($Emb_i$ for the first layer) and the shared demand and scenario-specific demands are extracted separately as $E_{share}^l$ and $E_i^l$ by experts and then are combined through a gating network as the output of layer-$l$, namely the demand representation $D_i^l$. The structure of the gating network of scenario $i$ in layer $l$, denoted as $g_{di}^l(\cdot)$, is based on a single-layer feed-forward network with Softmax as the activation function, taking input as the selector to generate the weighted sum of the output of experts. More precisely, the output of scenario $i$'s gating network within $l$-th extraction layer is formulated as :

$$D_i^l = g_{di}^l(D_i^{l-1})\text{Concat}(E_i^l, E_{share}^l), \tag{1}$$

where $g_{di}^l(D_i^{l-1})$ denotes a gating network to calculate the scenario $i$'s weight vector as:

$$g_{di}^l(D_i^{l-1}) = \text{Softmax}(W_i^l D_i^{l-1}), \tag{2}$$

where $W_i^l \in \mathbb{R}^{m \times d}$ is a parameter matrix, with dimension $m$ as the number of experts and $d$ as the length of input representation $D_i^{l-1}$, particularly, $D_i^0$ is exactly $Emb_{di}$.

• **Price Competitiveness Representation Extraction (PCRE)** In this module, scenario-wise price competitiveness is separately captured by attention mechanism [4]. Considering that even under the same scenario, the price competitiveness of different listing contributes to each other in a complex way. To model this pattern, We propose PCRE module that is composed with **Price Competitiveness Extraction (PCE)** and **Price Competitiveness Integration (PCI)**. Firstly, PCE decomposes price competitiveness into three parts: **internal competitiveness** (competitiveness among similar competing hotels at the same platform), **external competitiveness** (compared to same-hotel prices on other platforms within recent 30 days) and **self competitiveness** (compare current price to recent 30-day same-listing prices to capture temporal demand fluctuation). Sub-module $PCE^I$, $PCE^E$ and $PCE^S$ of $PCE$ are designed based on attention networks to learn the three competitiveness, denotes as $C_i^I$, $C_i^E$, $C_i^S$ respectively. After that, PCI integrates the three abstracted competitiveness representation, denoted as $C_i'$ through a gating network. All the following demonstration takes target listing $j$ in scenario $i$ as an example, thus for brevity, we omit the notation $i$ and $j$ in these formulas.

$PCE^I$ is an attention module, designed to capture the internal competitiveness. More precisely, it learns the competitiveness of the current listing price among a group of competing hotels, where hotels have similar qualities. Thus, they compete with each other in the shared market. Inspired by [34], to identify the competing room groups, we apply K-means clustering algorithm to cluster the rooms into K subgroups (K equals to the number of business districts) according to profile features such as brand, rating, location, etc. Based on this, $PCE_I$ takes the embedding of hotel profile as well as the counterpart of 5 pre-selected similar hotels as input to calculate the importance weights of similar hotels to the current

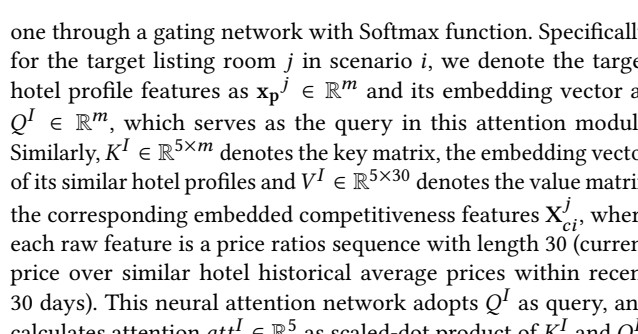

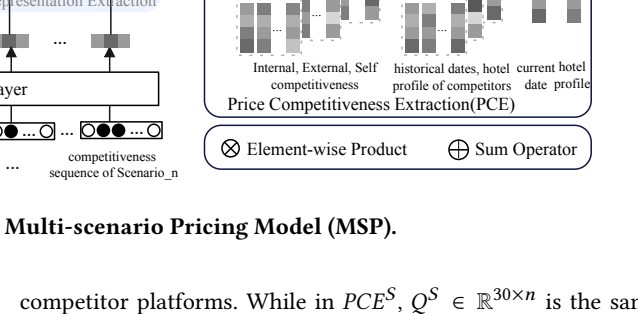

**Figure 2: The Architecture of Proposed Multi-scenario Pricing Model (MSP).**

one through a gating network with Softmax function. Specifically, for the target listing room $j$ in scenario $i$, we denote the target hotel profile features as $\mathbf{x_p}^j \in \mathbb{R}^m$ and its embedding vector as $Q^I \in \mathbb{R}^m$, which serves as the query in this attention module. Similarly, $K^I \in \mathbb{R}^{5 \times m}$ denotes the key matrix, the embedding vector of its similar hotel profiles and $V^I \in \mathbb{R}^{5 \times 30}$ denotes the value matrix, the corresponding embedded competitiveness features $\mathbf{X}_{ci}^j$, where each raw feature is a price ratios sequence with length 30 (current price over similar hotel historical average prices within recent 30 days). This neural attention network adopts $Q^I$ as query, and calculates attention $att^I \in \mathbb{R}^5$ as scaled-dot product of $K^I$ and $Q^I$:

$$att^I = \frac{K^I Q^I}{\sqrt{m}}. \tag{3}$$

The final output of $PCE^I$ is the weighted sum of values $V^I$, represented as $C_i^I$ for scenario $i$:

$$C_i^I = \sum_k^5 \frac{e^{att_k^I}}{\sum_j^5 e^{att_j^I}} * V_k^I. \tag{4}$$

Similarly, the output of sub attention modules $PCE^E$ and $PCE^S$ are calculated as:

$$
\begin{aligned}
C_i^E &= \sum_k^{30} \frac{e^{att_k^E}}{\sum_j^{30} e^{att_j^E}} * V_k^E, \\
C_i^S &= \sum_k^{30} \frac{e^{att_k^S}}{\sum_j^{30} e^{att_j^S}} * V_k^S,
\end{aligned}
\tag{5}
$$

where in $PCE^E$, $Q^E \in \mathbb{R}^n$ denotes the embedding of today's contextual features $\mathbf{x_t} \in \mathbb{R}^n$ and $K^E \in \mathbb{R}^{30 \times n}$ denotes context feature embedding of nearest 30 days with prices of the same room from

competitor platforms. While in $PCE^S$, $Q^S \in \mathbb{R}^{30 \times n}$ is the same as $Q^E$ and $K_i^S \in \mathbb{R}^n$ is instead the context feature embedding of nearest 30 days with the price from our platform of the same room.

PCE then concatenate all three price competitiveness representations as $C_i = \text{Concat}(C_i^I, C_i^E, C_i^S)$ as the input of Price Competitiveness Integration (PCI) module.

Considering that PCE extracts price competitiveness representation separately and inspired by MoE [9, 14, 15, 24, 32], we design PCI to better integrate the three competitiveness representations in $C_i$ and extracts deeper information through multiple MLP structures with a Softmax gating network to calculate the importance wight, and finally outputs the weighted summation $C_i'$ as the synthetic price competitiveness representation of this listing in scenario $i$, which is formulated as:

$$C_i' = g_{ci}(C_i) S_i(C_i), \tag{6}$$

where $C_i$ is the concatenated price competitiveness representation as mention above, and $g_{ci}(\cdot)$ is a weighting function with the same structure as $g_{di}(\cdot)$ to calculate the weight vector of different experts (MLP) knowledge $E_k(C_i)$. Concatenation is denoted as $S_i(C_i)$:

$$S_i(C_i) = \text{Concat}(E_1(C_i), ..., E_k(C_i)). \tag{7}$$

*3.2.2 Price suggestion.* Price adjustment ratio $V$ is then derived from the good-day probability $q$ according to a non-linear function. Several assumptions are behind this function:

(1) Price adjustment ratio $V$ is positively correlated with current good-day probability $q$.
(2) Price adjustment ratio $V$ is centered around 1, such that the suggested price always falls within a reasonable upper and lower range of the current price where we believe the theoretical optimal price lies.

Based on the assumptions, we introduce an adjustment function as $V = 1 + \alpha q^{A-q} - \beta$, where $\alpha$ controls the increasing magnitude. It falls between 0 and 1 to ensure that $V$ monotonically increases with $q$. $\beta$ controls the decreasing magnitude and keeps $V$ positive. A is a hyper-parameter that shapes the curve bend of the non-linear relation between price adjustment rate $V$ and good-day probability $q$. In our MSP, after extracting demand representation $D_i$ and price competitiveness $C_i'$ through DRE and PCRE, tower modules then learn the mapping from extracted information to good-day probability $q$ as well as parameters $\alpha$ and $\beta$ of the price suggestion function. Specially, to enhance the model awareness of more real-time price competitiveness information, $tower_\alpha$ and $tower_\beta$ takes $C_i'$ solely as input. Finally, the suggested price adjustment ratio $v_i^j$ of listing $j$ in scenario $i$ can be calculated as:

$$
\begin{aligned}
q_i &= t_i^q(\text{Concat}(D_i, C_i')), \\
\alpha_i &= t_i^\alpha(C_i'), \\
\beta_i &= t_i^\beta(C_i'), \\
v_i^j &= 1 + \alpha_i q_i^{A-q_i} - \beta_i,
\end{aligned}
\tag{8}
$$

where $t_i^q(\cdot)$, $t_i^\alpha(\cdot)$, $t_i^\beta(\cdot)$ denotes the tower networks of scenario $i$.

*3.2.3 Training objective.* The final loss function consists of two parts, good-day probability $loss_{qi}$ and price suggestion $loss_{vi}$ for each scenario $i$ [31] as follows:

$$
\begin{aligned}
loss_i &= w_q \sum_i loss_{qi} + w_v \sum_i loss_{vi}, \\
loss_{qi} &= -\sum_j y_i^j * \log \hat{q}_i^j, \\
loss_{vi} &= \sum_j (\max(0, L(y_i^j) - V) + \max(0, V - U(y_i^j))),
\end{aligned}
\tag{9}
$$

where $loss_{qi}$ is the cross-entropy loss of our good-day probability prediction $q_i^j$ and good-day label $y_i^j$ in scenario $i$, and $loss_{vi}$ mainly measures the gap between our suggested price and the theoretical optimal price. Besides, $w_q$ and $w_v$ are hyper-parameters that adjust the ratio of the two loss.

Note that in our case, the exact optimal price is unavailable. Based on assumption 2, we introduce hyper-parameters $c_1$ and $c_2$ to control the range $[L(y_i^j), U(y_i^j)]$ where we believe the optimal price adjustment ratio lies in between. Correspondingly, $L(y_i^j)$ denotes the lower bound and $U(y_i^j)$ denotes the upper bound of the optimal price adjustment ratio based on the current calendar price. They can express as follows:

$$
\begin{aligned}
L(y_i^j) &= y_i^j + (1 - y_i^j) * c_1, \\
U(y_i^j) &= y_i^j * c_2 + (1 - y_i^j).
\end{aligned}
\tag{10}
$$

For those good-day samples, $L(y_i^j)$ is exactly 1 to prevent a price suggestion from decreasing, while the upper bound is blurry but constrained by $c_2$. In the counterpart of bad-day samples, $L(y_i^j)$ is higher than $c_1$ and $U(y_i^j)$ is the current price. Since $c_1, c_2$ are closely connected to business logic, according to our platform pricing needs, we set them as 80% and 120%. If our suggested price falls between

**Table 1: Occupancy and Suggest Price Cases Summary.**

| Occupancy | Optimal Price | Suggested Price | Result |
|---|---|---|---|
| Good Day | $P_o \geq P$ | $P > P_{sug}$ | bad |
| Good Day | $P_o \geq P$ | $P \leq P_{sug}$ | good |
| Bad Day | $P_o < P$ | $P \leq P_{sug}$ | bad |
| Bad Day | $P_o < P$ | $P > P_{sug}$ | good |

**Table 2: Number of Samples in Each Four Cases.**

| | Good Day | Bad Day |
|---|---|---|
| $P_{sug} \geq P$ | a | b |
| $P_{sug} \leq P$ | c | d |

$L(y_i^j)$ and $U(y_i^j)$, the $loss_v^i$ is 0, otherwise, it is the distance between the suggested price and the nearer bound.

When the model performs well according to $loss_{qi}$, it makes sure the validity when the problem is reduced to a binary classification problem of whether to reduce prices. In this case, our pricing suggestion is at least in the right direction, namely suggesting price increase for good-day listings while suggesting price decrease for bad-day listings. As for $loss_{vi}$, given the right pricing direction, $loss_{vi}$ keeps the final suggested price within a reasonable range to the current price. So we set $w_q$ to 1 while limit $w_v$ within range $(0,1)$, and finally set $w_v$ as 0.1 after hyper-parameters experiments.

## 4 PERFORMANCE EVALUATION

We propose a metric scheme that is compatible with different sample distributions. In this way, a well-rated pricing strategy can reach a balance between reducing the price of those listings with occupancy not living up to expectations, and raising the price of the already good ones. Refinements are based on the traditional evaluation framework proposed by Ye et al. [31], which is widely adopted [25, 33]. Let's denote the current price as $P$, our suggested price as $P_{sug}$, and the ideally optimal price as $P_o$. As defined before, we regard it as a good day if actual occupancy is higher than or equal to the threshold for each listing under a specific scenario. Intuitively, the optimal price $P_o$ should be higher than or equal to the current price $P$ for those good days, and $P_o$ should be lower than $P$ to boost sales for those bad days. We regard $P_{sug}$ as bad when $P_{sug}$ is lower than $P$ for a good-day listing and when $P_{sug}$ is higher than or equal to $P$ for a bad-day listing. All cases with different listing occupancy and suggest prices are summarized in Table 1.

Assuming the number of suggestions in each case is defined in Table 2, we define a set of metrics as below:

- **Price Decrease Recall (PDR)**: among all bad-day listings, the percentage of price-decreasing suggestions.

$$
PDR = \frac{d}{b+d}
\tag{11}
$$

- **Price Increase Recall (PIR)**: among all good-day listings, the percentage of price-increasing suggestions.

$$
PIR = \frac{a}{a+c}
\tag{12}
$$

- **Booking Regret (BR)**:

$$BR = median_{goodday}\left(max\left(0, \frac{P - P_{sug}}{P}\right)\right) \quad (13)$$

- **Non-booking Regret (NR)**:

$$NR = median_{badday}\left(max\left(0, \frac{P_{sug} - P}{P}\right)\right) \quad (14)$$

As the traditional metric proposed by Airbnb [31], PDR and BR are mainly considered. PDR is used as the main metric and BR serves as the auxiliary. In particular, PDR measures how likely our price suggestion is lower than the current listing price for a bad-day listing. BR measures how close our suggested prices are to the booked prices when we suggest decreasing. This reflects how closely the suggested price aligns with the actual price and indicates how bad the bad cases are among those good-day listings, thus a pricing strategy with lower BR is more credible. As drawn by Airbnb, there is a trade-off between the PDR and BR according to experiment results. It is intuitive since, for a broadly decreasing strategy, the suggested prices go further away from the calendar prices, which leads to both increasing PDR and BR.

Both PDR and BR are highly correlated with the booking gain but partially and with bias. In real-world data sets, class imbalance often occurs and the distribution of positive and negative samples would change with time. The traditional metrics can't handle this since they solely focus on decreasing pricing suggestions. Let's consider a strategy that prefers to decrease price overall, while it helps to improve the competitiveness of those bad-day listings, it also hurts the benefit of those good-day ones. Taking an extreme example, a strategy that decreases all prices by 1% can cheat on the traditional metrics with PDR=100% and BR=1%. When our samples are all bad-day listings, it is acceptable, however, as the proportion of good-day listings increases, such a 'good-rated' pricing suggestion can be a nightmare. Apparently, such metrics focus more on decreasing pricing suggestions so that lead to biased pricing strategy.

To tackle this, we proposed $PRF_w$ and $BR_w$ to better accommodate the uneven sample. Our proposed metrics measure both price increasing and reduction suggestions comprehensively and adjust attention weight according to sample distribution. Similarly, $PRF_w$ is mainly considered while $BR_w$ serves as auxiliary.

$$PRF_w = \frac{PDR * PIR}{wPDR + (1 - w)PIR} \quad (15)$$

$$BR_w = max(BR, NR) \quad (16)$$

where $w$ is the average ratio of good-day listings according to the sample. In the extreme cases, our weighted metrics will be as in Table 3. Let's consider again the extreme decreasing strategy mentioned above. According to our metrics, when all samples are good-day listings, the simple biased strategy earns scores as $PRF_w = 0$ and $BR_w = 1\%$, which is defeated by the random strategy and is easily ruled out.

## 5 EXPERIMENTS

In this section, we conduct extensive offline experiments and online A/B tests to evaluate the effectiveness of the proposed MSP. We mainly focus on the following research questions:

**Table 3: Metric Score under Extreme Circumstance.**

| Extreme Case | $PRF_w$ | $BR_w$ |
|---|---|---|
| All Good (w=1) | PIR | $BR_w$ |
| All bad (w=0) | PDR | $BR_w$ |
| Random Strategy | 0.5 | $BR_w$ |
| Ideal Strategy | 1 | 0 |

**Table 4: Statistics of Multi-scenario Hotel Reservation Dataset.**

| Dataset | Scenario | # deals | deal ratio |
|---|---|---|---|
| Training | A | 327,337 | 0.2526 |
| 2023.06.01 - | B | 559,915 | 0.4321 |
| 2023.06.30 | C | 506,709 | 0.3911 |
| Testing | A | 90,142 | 0.2648 |
| 2023.07.01 - | B | 148,483 | 0.4363 |
| 2023.07.07 | C | 126,711 | 0.3723 |

- **RQ1**: How does MSP perform on the hotel pricing task compared to baseline methods?
- **RQ2**: Where is the improvement in multi-scenario pricing strategy compared to single scenario pricing method?
- **RQ3**: How do different modules of MSP contribute to the model performance?
- **RQ4**: How does our proposed MSP perform on the platform's live production environment?

### 5.1 Experiment Setup

*5.1.1 Dataset. Single-scenario Dataset.* Offline experiments are conducted on hotel pricing datasets Dataset-H and Dataset-L constructed in [34] to verify that the multi-scenario pricing model has stable performance on single-scenario pricing tasks. Dataset-H contains hotel reservation data at platform A during 2020.12.17 - 2021.01.23 (high-booking season) and Dataset-L contains reservation data during 2021.03.11 - 2021.04.17 (low-booking season).

*Multi-scenario Dataset.* As there is no public dataset for multi-scenario hotel pricing, we construct the first real-world multi-scenario pricing dataset based on hotel transaction data of 56,949 hotels, 105,102 rooms and 59 check-in dates at platform A from 2023.06.01 to 2023.07.07, providing different pricing scenario based on three distribution channel, which is released publicly with this paper. This multi-scenario hotel pricing dataset contains already constructed features and corresponding label, details of which are summarized in Table 4. In our experiment, first 30-day data is used to train the model, and the left 7-day data is used for testing.

*5.1.2 Baseline Methods.* We compare our MSP with following baselines. Since all the baseline methods are originally proposed for single-scenario pricing tasks, we treat pricing tasks under different scenarios as independent and train baseline models for each.

- *Meituan pricing* [33] adopts a DNN model to regress the suggested price for a certain room at a specific date.
- *Airbnb pricing* [31] uses a customized regression model to suggest the price for Airbnb listing-nights and applies personalized logic to optimize the suggestion.


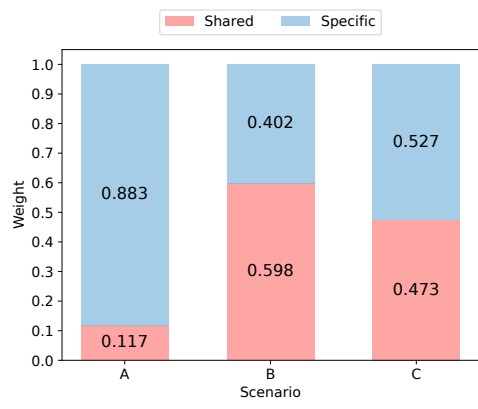

Figure 3: Experts Utilization in Different Scenarios.

• *PEM pricing* [34] determines the suggested price by introducing an elastic demand-based function that captures the price elasticity of demand in the hotel purchase prediction model.

Table 5: Performance Comparison on Single-scenario Pricing.

| Method | Dataset-H | | | | Dataset-L | | | |
|---|---|---|---|---|---|---|---|---|
| | $PDR$ | $PIR$ | $PRF_w$ | $BR_w$ | $PDR$ | $PIR$ | $PRF_w$ | $BR_w$ |
| DNN | 54.0 | 27.8 | 42.9 | 7.8 | 60.6 | 35.9 | 53.3 | 5.6 |
| Airbnb | 54.8 | 27.0 | 42.7 | 8.1 | 60.3 | 34.3 | 52.4 | 5.5 |
| PEM | 56.2 | 30.1 | 45.4 | **8.5** | 61.3 | 37.9 | 54.6 | **6.0** |
| MSP-single | 56.9 | 30.4 | **45.9** | 8.3 | 62.1 | 37.8 | **55.1** | 5.6 |

*5.1.3  Implementation details.* In MSP, the embedding size of all features is set to 8, and the number of units in the GRU structure is set to 128. The DRE Module contains two information abstraction layers, and each task contains 3 experts with 128 units. The PCE Module shifts attention over 5 similar hotels and context features of the past 30 days. Each MLP deployed in PCI has 2 hidden layers of sizes 256 and 128. In the loss function, $w_q = 1$, $w_v = 0.1$, $c_1 = 0.8$ and $c_2 = 1.2$. MSP is trained through the Adam optimizer with a learning rate of 0.001 and a batch size of 512. For the baseline methods, scenarios are treated as independent and, in order to achieve a fair comparison, we keep internal structure complexity of the baseline models consistent with MSP.

## 5.2  Offline Experiment

*5.2.1  Performance Comparison (RQ1).* The performance of all models in Comparison Experiments is summarized in Table 5 and Table 6 in percentage. We can draw the following conclusion:

(1) **Effective pricing strategy** Our proposed MSP outperforms all the baseline models on the $PRF_w$ metric both on the single-scenario dataset and the newly released multi-scenario dataset. It gains an improvement of 20.1% in scenario C, 6.3% in scenario B and 0.8% in scenario A, compared to the best-performed baseline models respectively. $BR_w$ measures how close the suggested prices are to the actual prices when the pricing suggestion diverges from the optimal strategy. From this viewpoint, our MSP model wins in scenarios C and B by 57% and 46% reduction of $BR_w$. It's worth noting that, single scenario version MSP (MSP-single) that solely takes data from one scenario as input still gains improvements compared to the best-performed baseline model in each scenario. This further validates the effectiveness of MSP pricing strategy and its compatibility with single- and multi-scenario pricing problems.

(2) **Advantageous information extraction ability** MSP makes impressive improvement in scenario C which distinguishes from the other two scenarios for pricing decisions are applied to distributor prices that won't be seen directly by customers. Apart from this, view/search/click data used to extract market demand is unavailable in this scenario. In this case, the single-scenario pricing model suffers to capture the demand and price competitiveness.

(3) **Comprehensive offline metrics** MSP is compared to random strategy and the extreme decreasing strategy mentioned in Section 4. According to the biased traditional metrics, the extreme decreasing strategy defeats any other methods with perfect $PDR = 1$ and considerably small $BR_w = 0.01$. However, it gains $PRF_w = 0$ in our newly proposed metrics and surely won't be chosen. This proves our proposed metrics is more comprehensive and can easily rule out irrationally biased pricing strategy.

*5.2.2  Expert Utilization Analysis (RQ2).* The DRE in our model leverages demand data across scenarios to tackle data sparseness. To disclose how scenario-specific and shared information are extracted and aggregated in different scenarios, we investigate expert utilization of those gate-based sub-modules. Figure 3 shows the weight distribution of experts utilized by each gate calculated on test data, where the height of bars are the average weight value. It shows the combinations of shared and specific experts varies significantly between 3 scenarios. Scenario B and C relies much more on shared experts than scenario A, which explains the significant performance promotion on pricing task in these two scenarios.

*5.2.3  Ablation Study (RQ3).* To analyze the effectiveness of our proposed sub-modules DRE and PCRE (PCI and PCE included) we design 4 variants in the ablation study:

• **MSP w/o DRE**: a variant of MSP which replaces DRE with the embedding of scenario-wise demand features;

• **MSP w/o PCRE**: a variant of MSP which deletes PCRE, and the input of DRE contains the embedding of competitiveness sequence and demand sequence;

• **MSP w/o PCI**: a variant of MSP which deletes PCI;

• **MSP w/o PCE**: a variant of MSP which deletes PCE;

Table 7 shows the overall experiment results in persentage. Our proposed MSP dominates any other variants according to $PRF_w$ with $BR_W$ controlled in an acceptable range, which validates the effectiveness of MSP. It's worth noting that MSP w/o PCRE has a serious performance degradation in scenario A with $PRF_w = 0.0001$ and $BR_W = 0.1777$, while it performs stable in scenario B and achieves considerable improvement in scenario C. The reason is that, scenario A, as it has the most comprehensive demand data and competitive price information, benefits least from the across-scenario demand sharing structure DRE, and relies more on price competitiveness information extracted by PCRE. Thus the removal of PCRE reduces the model effectiveness greatly for scenario A, while for scenario B and C, it is the opposite. This is clearly a domain seesaw phenomenon, the performance of some scenarios get improved at the cost of hurting the performance of others.

**Table 6: Performance Comparison on Multi-scenario Pricing.**

| Method | Scenario A | | | | Scenario B | | | | Scenario C | | | |
|---|---|---|---|---|---|---|---|---|---|---|---|---|
| | $PDR\uparrow$ | $PIR\uparrow$ | $PRF_w\uparrow$ | $BR_w\downarrow$ | $PDR\uparrow$ | $PIR\uparrow$ | $PRF_w\uparrow$ | $BR_w\downarrow$ | $PDR\uparrow$ | $PIR\uparrow$ | $PRF_w\uparrow$ | $BR_w\downarrow$ |
| random | 50.0 | 50.0 | 50.0 | \ | 50.0 | 50.0 | 50.0 | \ | 50.0 | 50.0 | 50.0 | \ |
| extreme decrease | 100.0 | 0.0 | 0.0 | 1.0 | 100.0 | 0.0 | 0.0 | 1.0 | 100.0 | 0.0 | 0.0 | 1.0 |
| Meituan | 60.2 | 41.0 | 55.3 | **4.8** | 26.9 | 78.7 | 31.9 | 42.7 | 90.3 | 13.1 | 40.4 | 22.6 |
| Airbnb | 60.9 | 40.3 | 55.5 | 14.5 | 49.0 | 53.0 | 49.9 | 4.4 | 91.9 | 22.7 | 56.2 | 18.9 |
| PEM | 61.1 | 41.0 | 55.8 | 15.3 | 51.3 | 55.5 | 52.3 | 7.5 | 90.4 | 24.0 | 57.2 | 20.4 |
| MSP-single | 61.5 | 40.6 | 55.9 | 12.6 | 57.7 | 46.3 | 54.5 | 4.2 | 77.6 | 34.0 | 61.2 | 9.2 |
| MSP | 62.8 | 39.3 | **56.3** | 7.0 | 59.6 | 45.6 | **55.5** | 2.3 | 82.2 | 42.4 | **68.7** | 3.8 |

**Table 7: Ablation Study Results.**

| Method | Scenario A | | | | Scenario B | | | | Scenario C | | | |
|---|---|---|---|---|---|---|---|---|---|---|---|---|
| | $PDR\uparrow$ | $PIR\uparrow$ | $PRF_w\uparrow$ | $BR_w\downarrow$ | $PDR\uparrow$ | $PIR\uparrow$ | $PRF_w\uparrow$ | $BR_w\downarrow$ | $PDR\uparrow$ | $PIR\uparrow$ | $PRF_w\uparrow$ | $BR_w\downarrow$ |
| MSP w/o DRE | 60.9 | 41.2 | 55.7 | 13.5 | 54.6 | 46.9 | 52.6 | 6.0 | 91.5 | 25.3 | 59.1 | 19.8 |
| MSP w/o PCRE | 99.9 | 0.0 | 0.0 | 17.7 | 54.9 | 51.5 | 54.0 | **0.1** | 84.9 | 36.4 | 66.3 | 9.9 |
| MSP w/o PCI | 62.5 | 38.2 | 55.7 | 21.1 | 59.5 | 41.6 | 54.0 | 4.3 | 93.6 | 17.2 | 48.5 | 13.2 |
| MSP w/o PCE | 60.7 | 41.0 | 55.6 | 15.9 | 60.1 | 41.3 | 54.3 | 8.8 | 91.1 | 24.2 | 57.8 | 14.3 |
| MSP | 62.8 | 39.3 | **56.3** | 7.0 | 59.6 | 45.6 | **55.5** | 2.3 | 82.2 | 42.4 | **68.7** | 3.8 |

The experiment results verifies the effectiveness of MSP, which explicitly separates shared demand information and independent competitiveness information extraction process through DRE and PCRE modules. Such structure can well prevent from negative transfer by not introducing unnecessarily noisy information to scenario A with sufficient data itself while achieves information sharing for data-sparse scenarios B and C. In addition, variants MSP w/o PCI and MSP w/o PCE validate the effect of sub-module PCI and PCE of PCRE in each scenario. Apart from this, for MSP w/o DRE, $PRF_w$ of scenario A/B/C drops by 1.0%, 5.3% and 14.0% respectively, which suggests that the shared demand information extracted by DRE can effectively complement the specific demand representation especially for scenario B and C. Superior performance of full MSP structure over MSP w/o PCRE and MSP w/o DRE demonstrates that the explicitly separated extraction structure of scenario-shared and scenario-independent information in MSP achieves a balance between leveraging data and avoiding domain seesaw.

### 5.3 Online A/B test (RQ4)

To further validate the online performance of the proposed MSP as well as our proposed offline metrics, we conducted a four-week online A/B test at platform A against baseline model PEM, one of the state-of-the-art models that outperforms others in the offline experiments. Figure 4 shows that MSP averagely achieves 1.4%, 1.9% and 7.3% relative improvement in GMV compared to PEM. In accord with offline experiment results, scenario C improves most among the three scenarios by adopting MSP model, which further demonstrates that MSP leverages data across scenarios and avoid domain seesaw in addressing the multi-scenario hotel pricing problem at OTPs. Besides, let's consider again the extreme decreasing strategy mentioned in Section 4 that would dominate any other baseline models according to traditional metrics. Such a broad but minor decrease-pricing strategy, in fact, won't cause any fluctuation on GMV revenue in the online experiment, and thus is apparently inferior to MSP strategy according to the online experiment results. This proves the validity of our newly proposed

offline metrics by demonstrating it is more closely linked to online business growth.

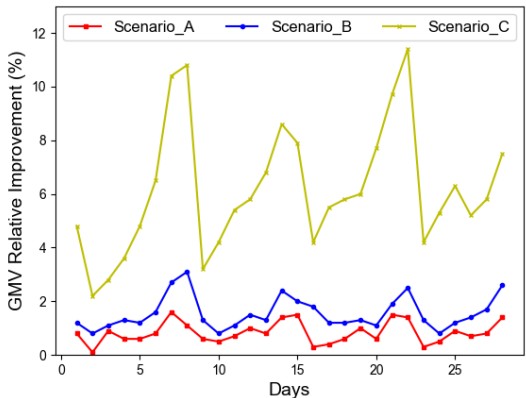

**Figure 4: Online A/B Test Results.**

### 6 CONCLUSION

Previous studies have delved into dynamic pricing strategy and disclosed that pricing plays a crucial role in user's purchase behavior and platform revenue management. However, existing methods mainly focus on optimizing prices in a specific scenario, which leads to unsatisfactory performance. In this paper, we conduct in-depth analysis based on industrial transaction data from platform A and propose MSP, which leverages the multi-scenario data and optimize price suggestions in different scenarios. On this basis, the DRE and PCRE modules capture sharing demand and scenario-specific price competitiveness separately to enhance the representation learning of scenarios and avoid domain seesaw. The effectiveness of MSP is well validated through offline and online experiments, demonstrating the superiority over state-of-the-art baseline models. MSP is currently fully deployed in three scenarios at platform A and continually gains revenue promotion.

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
