# OpenReview forum: "Multi-Scenario Pricing for Hotel Revenue Management"
_ACM.org/TheWebConf/2024/Conference — TheWebConf24 Oral_

### Official Review · Reviewer_kkZ9 · 2023-11-21

**Novelty:** 5
**Technical Quality:** 5

**Review:**

This paper presents a Multi Scenario Pricing (MSP) model, which incorporates shared information across multiple scenarios. The results of offline and online experiments on real-world datasets demonstrate the superiority of MSP in terms of performance.

Strengths:

1) This paper demonstrates a well-structured and clearly articulated methodology. Additionally, the availability of real-industry multi-scenario pricing data and code further strengthens the arguments presented in this study.

2) This paper introduces a novel pricing metric that comprehensively measures both price increases and reductions. It can shift the attention to sample distribution from a lower price.

Weaknesses:

1) The detailed process of the MSP method lacks clarity. Formal summarization of the steps and execution process of the MSP method in the form of Algorithm in section 3.2 would enhance the clarity.

2) This paper lacks convergence curve plots for loss during the training process, and it is necessary to analyze the convergence speed and stability of the model in section 5. Furthermore, providing a comparison curve between the model's suggested price and theoretical optimal price would offer more intuitive insights.

**Questions:**

1) Can the dynamic pricing considering shared information between different scenarios also be approached from a spatiotemporal modeling perspective? By considering temporal and spatial changes, comprehensively and accurately understanding the shared information among different scenarios may enhance the effectiveness of pricing strategies.

2) In section 5.3, you just show the relative performance improvements of MSP in online A/B test. Can you show the absolute performance metrics of MSP and PEM?

3) The multi-scenario industry dataset in this paper is the first real-industry multi-scenario pricing data, but the statistics of this dataset is too single. Could you consider adding some more detailed introduction and analysis to the dataset?

**Ethics Review Description:**

-

**Reviewer Confidence:**

3: The reviewer is confident but not certain that the evaluation is correct

**Scope:**

3: The work is somewhat relevant to the Web and to the track, and is of narrow interest to a sub-community

---

### Official Review · Reviewer_r7dj · 2023-11-22

**Novelty:** 5
**Technical Quality:** 5

**Review:**

1. Summary

The paper proposes a new dynamic pricing method for hotels, the multi-scenario pricing (MSP) model, which utilizes cross-scenario and specific information to capture more accurate market demand and competitiveness. The MSP model clearly separates information into shared components and scenario-specific price competitiveness to prevent domain swings. The paper proposes a more comprehensive pricing metric that adjusts attention based on sample distribution to accommodate category imbalance, which is closely related to online business metrics. The work also provides a large-scale production data set collected from an online travel platform and is the first industrial production data set with multi-scenario pricing.

2. Strengths and Weaknesses

Strength:

1)	This paper proposes more comprehensive model evaluation indicators that shift attention based on sample distribution, adapt to the imbalance of data categories, and can more reasonably evaluate the quality of the model.

2)	This paper conducted multiple categories of experiments on the MSP model and explained its superiority through extensive experiments and analysis. The experiment in the production environment further proves the effectiveness of the MSP model.

3)	An industrial-level dataset is released.

Weakness:

1)	The major concern is about the limited baselines selected for the experiments. Only three methods for a single scenario are used as baselines. The results would be more convincing if models for mult-scenarios could be added as baselines.

2)	There is a limited description of related works for the multi-scenario pricing, which is important for readers to know what efforts have been made previously in this setting.

**Questions:**

1.	In Table 6, the BRw index of the Meituan method in three scenarios has a large gap. It is the lowest in scenario A but the highest in scenarios B and C. What is the reason for this?

2.	In Table 5, the first baseline method is DNN, but in Table 6, the first baseline method is Meituan. Are these two methods the same?

**Reviewer Confidence:**

3: The reviewer is confident but not certain that the evaluation is correct

**Scope:**

4: The work is relevant to the Web and to the track, and is of broad interest to the community

---

### Official Review · Reviewer_vd9Y · 2023-11-22

**Novelty:** 5
**Technical Quality:** 5

**Review:**

This work develops a multi-scenario pricing algorithm. IMO, as a non-expert, this is a high-quality paper with clear explanations, visualization, analysis, and evaluation. I do not have a major comment, just some suggestions that the authors may want to consider. It is a good and decent paper. I specifically, as a non-expert, liked the evaluation methods.

-- Limited audience. It seems that this paper is written for a very specific audience in mind. There is nothing inherently problematic with that; however, for a conference with a broad audience, it is fair to say that this work does not attract many people. A discussion on the potential implications for the broader audience might be constructive.

-- One element that is missing is the confidence intervals. Are the claims statistically significant? I can see the point estimations show an improvement compared to the baseline and other models, but are they statistically significant? Estimating the confidence intervals can be an interesting research question. Confidence intervals are not just useful for model comparison. They also provide a measure to estimate risks associated with pricing and can be another evaluation factor. This can be a research question of its own.

**Questions:**

-- Are the claims statistically significant?

-- What can a general reader who is not interested in this topic learn from the proposed method? Does it have potential applications besides what is presented in the paper?

-- A discussion on potential fairness and bias implications can be useful. This paper is mainly concerned with maximizing the efficiency of firms. It would be nice to remind the readers that efficiency maximization without harm mitigation can impact individuals and/or groups in a negative way. For example, does it equally benefit all neighborhoods, or is the increase in performance unequally distributed to a specific group of people? I understand doing this analysis is another work by itself, but they should be acknowledged.

**Ethics Review Description:**

See questions

**Ethics Review Flag:**

Yes

**Reviewer Confidence:**

1: The reviewer's evaluation is an educated guess

**Scope:**

3: The work is somewhat relevant to the Web and to the track, and is of narrow interest to a sub-community

---

### Official Review · Reviewer_ie5F · 2023-11-23

**Novelty:** 6
**Technical Quality:** 5

**Review:**

The authors of the paper “Multi-Scenario Pricing for Hotel Revenue Management” present a novel neural architecture for the dynamic pricing task. The architecture is intended to tackle multiple pricing scenarios and is composed of several components which are “Demand Representation Extraction” and “Price Competitiveness Representation Extraction”. The authors support their architecture with extensive experimentation in both offline and online evaluations. Finally, 2 additional performance measures are suggested to reduce the chance of biasing the pricing strategy (to favor decreasing prices).

In my opinion the work is very interesting and the fact that one model handles all possible scenarios at once could provide high benefits to relevant stakeholders. With that being said, I do have some remarks regarding the presentation of the work. First, the introduction doesn’t clearly portray the different scenarios (A/B/C), it seems that their presentation might have been removed at some point. In addition, it seems that MSP does particularly when compared via the novel suggested metrics, however, when compared through PIR its performance is questionable. Finally, in the ablation tests, in some cases the results are very close with respect to MSP and the different variants, are the differences statistically significant?

All in all, although I have some remarks, I do believe that the proposed method is novel and interesting. Moreover, it was also tested in an online manner and provided positive results.

**Questions:**

Questions raised within the review.

**Ethics Review Description:**

.

**Reviewer Confidence:**

2: The reviewer is willing to defend the evaluation, but it is likely that the reviewer did not understand parts of the paper

**Scope:**

4: The work is relevant to the Web and to the track, and is of broad interest to the community

---

### Official Review · Reviewer_MNoG · 2023-11-24

**Novelty:** 5
**Technical Quality:** 5

**Review:**

This paper proposes a multi-scenario pricing scheme for hotel revenue management. One contribution is claimed to utilize shared information between different scenarios. As a contrast, existing pricing strategies are tailored to each specific scenario using data only from that scenario. The proposed approach called MSP leverages cross-scenario and specific information to capture more accurate market demand and competitiveness. More specifically, the model structure explicitly separates information into shared components as market demand and specific information as scenario-wise price competitiveness to prevent domain seesaw. To capture the inherent correlation between listings in different scenarios, an attention network named PCRE is designed. Another contribution is claimed to be the first real-industry multi-scenario pricing data. Extensive experiments are conducted to show the superior performance of the proposed approach. Overall, this paper is well-structured.

I have several concerns.

The key contribution of this paper is claimed to utilize shared information between different scenarios. It is not well motivated why capturing the shared information is so important and what are some key challenges in capturing this shared information. In the introduction, it only states that applying the dynamic pricing scheme designed for single scenario is challenging. This implies that applying this single scenario approach is not viable. But they does not show the underlying challenge of utilizing the shared information.

Point 3) in the second paragraph of Introduction may hold when the number of scenarios is large. When the number of scenarios is small, designing a pricing strategy for each scenario utilizing all data may yield a better performance and the cost is acceptable.

The third paragraph of Introduction is not well connected to the challenge analysis in the second paragraph of Introduction. Specifically, the third paragraph does not match challenge analysis in the second paragraph well.

The related work needs improvement. The technical difference is not clearly stated. For example, the difference to model based approaches is stated as “these priors may not meet with the real-world scenarios”. This is really a vague statement without any meaningful technical insights. The difference to data-driven approaches is stated as “they all ignore the shared information across multiple scenarios”. This is also vague. Essentially, this paper applies neural network for dynamic pricing. Some previous works also use this technique. The authors should clearly state the technical difference. Otherwise, the technical novelty of this paper is unclear and capturing the shared information is not convincing to warrant a publication.

This paper applies several types of neural network functions. The design choice should be stated.

**Questions:**

Please refer to my concerns.

**Reviewer Confidence:**

2: The reviewer is willing to defend the evaluation, but it is likely that the reviewer did not understand parts of the paper

**Scope:**

4: The work is relevant to the Web and to the track, and is of broad interest to the community

---

### Decision · Program_Chairs · 2024-01-22

**Decision:**

Accept (Oral)

**Comment:**

The paper proposes a novel pricing algorithm that uses sharing structure to leverage cross-scenario and specific information for better dynamic pricing in online travel platforms. The reviewers are in consensus of the novelty and quality of the contribution. They agree that the methodology is well structured and clearly articulated and may provide high benefits to relevant stakeholders. They also praise that the industry-setting multi-scenario pricing data.